# Tracking Antimicrobial Resistant *E. coli* from Pigs on Farm to Pork at Slaughter

**DOI:** 10.3390/microorganisms10081485

**Published:** 2022-07-23

**Authors:** Rupert Bassitta, Hanna Kronfeld, Johann Bauer, Karin Schwaiger, Christina Hölzel

**Affiliations:** 1Former Department of Animal Hygiene, Technical University of Munich, Weihenstephaner Berg 3, 85354 Freising, Germany; rupert.bassitta@lra-rosenheim.de; 2Department for Animal Hygiene, Animal Health and Food Safety, Institute of Animal Breeding and Husbandry, Christian-Albrechts-University, Olshausenstr. 40, 24098 Kiel, Germany; choelzel@tierzucht.uni-kiel.de; 3Unit of Food Hygiene and Technology, Institute of Food Safety, Food Technology and Veterinary Public Health, University of Veterinary Medicine, Veterinärplatz 1, 1210 Vienna, Austria; karin.schwaiger@vetmeduni.ac.at

**Keywords:** *Escherichia coli*, foodborne pathogens, antimicrobial resistance, antibiotics, food chain, food safety, pig meat

## Abstract

Antimicrobial-resistant bacteria might be transferred via the foodchain. However, that risk is rarely tracked along different production steps, e.g., from pigs at farm to meat. To close that gap, we performed a prospective study in four conventional and two organic farms from the moment pigs entered the farm until meat sampling at slaughter. Antimicrobial use was recorded (0 to 11 agents). Antimicrobial susceptibility (AMS) against 26 antibiotics, including critically important substances, was tested by microdilution, and *tet*A-*tet*B-*sul*I-*sul*II-*str*A-*str*B-*bla*-CTXM-*qac*EΔ1 were included in PCR-genotyping. From 244 meat samples of 122 pigs, 54 samples (22.1%) from 45 animals were positive for *E. coli* (*n* = 198). MICs above the breakpoint/ECOFF occurred for all antibiotics except meropenem. One isolate from organic farming was markedly resistant against beta-lactams including fourth-generation cefalosporines. AMS patterns differed remarkably between isolates from one piece of meat, varying from monoresistance to 16-fold multiresistance. Amplicon-typing revealed high similarity between isolates at slaughter and on farm. Prior pig lots andeven the farmer might serve as reservoirs for *E. coli* isolated from meat at slaughter. However, AMS phenotyping and genotyping indicate that antimicrobial resistance in *E. coli* is highly dynamic, impairing reliable prediction of health risks from findings along the production chain.

## 1. Introduction

Antimicrobials are a very valuable tool—or even weapon—in human medicine, but also in the treatment of livestock, and substances overlap broadly between both fields of use. Thus, similar resistances might be selected and transferred from livestock to humans via consumption. In 2020, penicillins (278 t), tetracyclines (148 t), and sulfonamides (65 t) were used most frequently in livestock in Germany; the quantities dispensed for 3rd- and 4th-generation cephalosporins were 1.0 t and 0.3 t, respectively (https://www.bvl.bund.de/SharedDocs/Bilder/09_Presse/01_Bilder_Pressemitteilungen/Tabelle_%20Antibiotika-Abgabemengen_2011-2020_Print.html, accessed on 19 July 2022). However, the individual quantities of active ingredients cannot be assigned to individual animal species because the majority of active ingredients are approved for use in different animal species. Similar to antibiotic use, antibiotic resistance is also monitored. Numerous monitoring programs are being performed or have been implemented in Europe, but such projects are not designed to track the origin of the resistant strains, which are mostly sampled from animal products by random inspections or from diseased humans or animals [1]. In the last decades, a variety of studies has investigated antimicrobial resistance in primary production—mainly pigs, chickens, and cattle. While a vast amount of studies deal with the prevalence of resistant isolates within a total lot of isolates, considerably fewer studies relate the prevalence of resistant isolates to animals or primary products thereof, and even less do so for sample numbers above 200. Therefore, even if there is basically ample information on the current antimicrobial resistance situation worldwide and the problems associated with it in both human and veterinary medicine, very little is known about the actual transmission routes. From farm to fork, many sources of entry are conceivable along the entire production chain. The primary task for food control is to avoid the spread of pathogenic bacteria that are listed in (EC) No 2073/2005. However, antimicrobial resistance is not primarily a problem associated with virulent subtypes; on the contrary, antimicrobial treatment might even be contraindicated in virulent strains, such as EHEC [2]. Anyhow, antimicrobial resistance might cause health risks independent of the virulence features of a strain, since carriers might form a reservoir for genetic transfer in the gut. It has been shown that, after oral ingestion of resistant bacteria via pork, bacteria survived the gastrointestinal passage and were detectable in the feces for up to 14 days after ingestion [3,4]. As early as in the 1960s, Williams Smith described the transient colonization of the human digestive tract with resistant *E. coli* strains of animal origin after oral ingestion [5]. Later studies also confirmed that the human microbiota is entered by resistant bacteria through the oral ingestion of bacteria-contaminated animal foods [3,6]. The reservoir function of such bacteria is the reason why, irrespective of virulence, commensal *E. coli* are included in the German implementation of Directive 2003/99/EC.

No matter whether meat samples are taken at the slaughterhouse or after processing, samplings show that pork can be contaminated with phenotypically resistant *E. coli* [7,8,9,10]. However, only very few tracking studies actually exist [11], due to the complexity of studies that have to be carried out over relatively long periods of time, additionally requiring the consent of those responsible at various stages of food production (e.g., farms and slaughterhouse). In a valuable study, Burow et al. [12] followed piglets from birth until the end of fattening, but no meat samples were included. In order to trace antimicrobial-resistant bacteria from pigs to pork along the production chain, we performed a prospective cohort study with pigs in farms and at slaughter in order to track along different production steps.

## 2. Materials and Methods

### 2.1. Study Farms

The study was conducted in six pig farms, with each farm assigned an identification code (A6, A21, E5, H18, J20, and M13). Two farms (H18 and J20) were organic pig farms, the others operated in a conventional production mode. Farms categorized as “organic” fulfilled at least the legal regulations of (EU) No 834/2007, now repealed by (EU) 2018/848. Two of the pig farms also had piglet production in addition to fattening (E5 and M13); these two farms did not purchase any fattening pigs. The other farms purchased animals from one source. Table 1 gives an overview of the individual farms and their characteristics.

### 2.2. Antimicrobial Use

Antimicrobial use data were collected from the compulsory documents related to veterinary application and dispensing, as well as by involving the herd books. The documents contain information on the number and type of animals treated, type and quantity of the administered drug, diagnosis, type and duration of application, dose per animal and day, as well as waiting periods. The antimicrobial treatment was prescribed and decided by the attending veterinarian of the respective farms. The antimicrobial treatments were therefore not influenced or controlled by the authors and only recorded retrospectively.

### 2.3. Sampling Procedure

Pigs were sampled at three different slaughterhouses, since the organically reared pigs were slaughtered at two separate slaughterhouses. Samples at slaughter were taken at the end of the regular fattening period (100–115 days in conventional farming, 126–135 in organic farming). Samples of intestinal content and meat samples were taken during the regular slaughter process from 10 randomly selected animals (sentinel) per fattening run (*n* = 1–4 per farm, depending on the time of study entry). Intestines were transferred to metal surface and rectum content was transferred to sterile tubes by manual manipulation of the rectum. The cecum was ligated, resected by use of a metal scissor, and transferred to sample bags. In the lab, the cecum was opened using a scalpel and content was transferred to a sterile tube. Meats samples were taken from neck and belly with the help of a sterile punch applicator (approx. 0.5 × 0.5 cm). In addition, feces samples had been taken before slaughter, from the same sentinel animals on farm during the fattening period during spontaneous defecation. The start of fecal sampling varied by farm: In E5, sampling started at the time of weaning; in M13, sampling started when the pigs entered the early growing-finishing phase (and pen) with approx. 25 kg. In the farms that purchased pigs, sampling started at the time of stalling. The sentinel animals were sampled monthly for the entire duration of the fattening period. In addition, 5 of the 10 sentinel animals were initially randomly selected and then sampled at each antimicrobial treatment—at the beginning, during, and after the end of the treatment (day 0, 2, 4, and 10). On day 0, 4, and 10, also the farmers provided stool samples by self-sampling.

### 2.4. Sample Processing and Escherichia coli-Isolation and Identification

The feces samples were suspended in NaCl solution (1 g in 9 mL) and a dilution series was prepared. Punch samples of the meat were swirled with 10 mL NaCl solution for 15 min at 200 rpm, and a dilution series was prepared from the suspension obtained. From those dilution series, 0.1 mL was spatted onto Fluorocult agar (Fluorocult ECD-Agar, Merck, Darmstadt, Germany) and incubated for 24 h at 44 °C. Colony material was also streaked to Gassner agar (Gassner-Agar, Merck, Darmstadt, Germany) and ESBL agar (CHROMagar ESBL, Mast Diagnostica, Reinfeld, Germany). In order to select for diversity, isolates were described morphologically. From feces samples, four isolates where picked that differed in colony morphology. From meat samples, all colonies were picked for further investigations. All isolates were confirmed by biochemical standard tests partly included in the cultivation media (fluorescence from conversion of 4-methylumbelliferyl-b-D-glucuronide; acid production from lactose, negative oxidase test, positive indole reaction). Questionable isolates were further confirmed by means of a miniaturized identification method, using modified conventional and chromogenic substrates (BBL Crystal Enteric/Nonfermenter ID Kit, Becton Dickinson, Franklin Lakes, NJ, USA). Isolates with conspicuous resistance profiles (in particular resistance to 3rd + 4th generation cephalosporins including ESBL phenotype) were additionally identified by MALDI-TOF MS (Matrix-assisted Laser Desorption/Ionisation-Time-of-Flight Mass Spectrometry, Bruker, Billerica, MA, USA).

Isolates were cryopreserved at −80 °C.

### 2.5. Antimicrobial Susceptibility Testing

The testing of phenotypic antimicrobial resistance was performed by microdilution following EUCAST (https://eucast.org/ast_of_bacteria/mic_determination/, accessed on 19 July 2022). The cryopreserved *E. coli* isolates were spread out on sheep blood agar and incubated at 37 °C for 24 h. The reference strain *E. coli* DSM 1103 was included as a performance control and was repeatedly tested together with each tested lot of study isolates. For testing, a bacterial suspension with a turbidity level according to McFarland 0.5 was prepared. Fifty microliters of the suspension were added to two test tubes, each containing 13 mL Müller-Hinton broth, and mixed briefly. The cell density of this suspension was approximately 5 × 10^5^ cfu/mL. For testing, commercial, but client-adapted 96-well microtiter plates pre-coated with antibiotics were used. Each well of the plate was filled with 100 μL suspension with the aid of the Micronaut^®^ Sprint automatic dispenser. The plate was covered with foil, shaken for five minutes, and then incubated for 18–20 h at 37 °C. Turbidity was visually evaluated. Based on listed breakpoints according to EUCAST, isolates were classified into susceptible, intermediate, or resistant. For three of the investigated antimicrobials, no EUCAST breakpoint was defined for *E. coli*. In those cases, epidemiological cut-off values (ECOFF; EUCAST) were used to classify the microdilution results. Breakpoints and ECOFFS are shown in Table 2.

### 2.6. Detection of Resistance Genes

DNA was extracted from pure cultures using a slightly modified form of the Chelex 100 DNA extraction method (Yang et al., 2008). The total volume of the PCR reaction was 25 µL, containing 23 µL of a commercially available master mix (LightCycler 480 SYBR Green I Master, Roche, Germany) composed of 18.25 µL nuclease-free water, 2.5 µL PCR buffer, 1.5 µL MgCl_2_ (25 µM), 0.5 µL dNTP mix (10 µM), and 0.25 µL Taq-Polymerase (5 U/µL), to which 0.5 µL primer FW, 0.5 µL primer RV (25 µM, each), and 1 µL of template-DNA were added. An overview of the relevant resistance genes investigated can be found in Table 3. PCR was performed in a thermocycler (T3000, Biometra, Jena, Germany) and included the amplification conditions listed in Table 4.

### 2.7. ERIC-PCR

In order to derive taxonomic relationships, an ERIC- (Enterobacterial Repetitive Intergenic Consensus) PCR was performed on *E. coli* isolates from four of the six farms (E5, A21, A6 and J20). The total volume of ERIC-PCR was 25 µL per sample, using 1 µL of the template DNA, adjusted to 15.38 ng per µL. A commercial master mix (GoTaqGreen Master Mix, Promega, Madison, WI, USA), which contains two dyes, was used to perform the ERIC-PCR under the conditions listed in Table 5. The target gene, fragment size, and annealing temperature of the primers used for ERIC-PCR can be found in Table 6.

The evaluation of the band patterns generated in the ERIC-PCR was carried out using the software GelCompar II (Applied Maths, Sint-Martens-Latem, Belgium). Digital images of the band patterns of the agarose gels were imported into the software program, identification numbers were assigned, and finally the cluster-analysis was carried out. Gels were aligned to each other by help of a DNA ladder and one *E. coli* strain was run on all gels in order to assess reproducibility.

## 3. Results

### 3.1. Antimicrobial Use

All farms except H18 (organic) applied antibiotics during the study period, but not necessarily in study animals: J20 treated only three pigs, which were not included in the study. A6, A21, E5, and M13 treated study animals as well as other animals housed in the farm. Applied antibiotics are listed in Table 7 and Table 8. As explained in material and methods, all antimicrobials were applied due to local veterinarian’s prescription for health reasons; veterinarians were not directed by the authors in doing so.

### 3.2. Detection of E. coli

Of 130 animals included on farm, 122 could be tracked until slaughter, where 244 meat samples were taken. Of these, 54 meat samples (22.1%) from 45 animals (36.9%) were positive for *E. coli*. Concentrations were 3.1 × 10^1^ to 9.9 × 10^2^ cfu per cm^2^). In total, 198 *E. coli* were isolated: 134 from belly meat and 64 from neck meat. In order to allow back-tracking, all isolates per plate (maximum number: 31) were included into further analysis. In Table 9, prevalences are listed farmwise.

### 3.3. MIC50/MIC90-Values of Antibiotics

Due to the very different number of isolates per farm, percentages of AMR were not calculated. Instead, MIC 50/MIC90-values were calculated provided that, for the respective antibiotic, any clinically resistant isolate was observed in animals from that farm at slaughter. Table 10, Table 11, Table 12 and Table 13 list MIC50/MIC90-values of antimicrobials approved for use in humans and/or in livestock. To sum up, among the antibiotics for which EUCAST provides breakpoints, there was only one antibiotic for which clinical resistance was totally absent in all 198 isolates: meropenem. While a similar number of isolates had been included for farms A6 and E5, the spectrum of clinical resistances was much broader in farm E5, where 19 antibiotics from 9 classes were affected (penicillins, cefalosporines, and carbapenems treated as different classes), compared to A6, where clinical resistances affected 7 antibiotics from 5 classes. As in E5, clinical resistance affected 19 antibiotics from 9 classes in A21, while 9 antibiotics from 4 classes were affected in M13. No obvious correlation could be detected when comparing the applied antibiotics with those that were affected by antimicrobial resistance (Table 10, Table 11, Table 12 and Table 13).

Prevalence of resistance could not be compared between organic and conventional production, since only six isolates from organic farming were available at slaughter. One of these isolates was markedly resistant against beta-lactams (10 affected antibiotics—thereof 7 cefalosporines—from 4 classes).

### 3.4. Phenotypic Susceptibility Patterns within the Same Sample

Susceptibility patterns differed remarkably between isolates from the same piece of meat, as shown in Table 14. Eleven different patterns were found for 29 isolates. Twenty-six of these patterns had the common feature of beta-lactam resistance against ampicillin, piperacillin, and amoxicillin plus clavulanate. Two isolates were sensitive for beta-lactams (but insusceptible to doxycycline), 22 isolates combined both patterns (doxycycline plus aminopenicillins) plus/minus further resistances. One isolate had an ESBL-phenotype with a broad range of affected cefalosporines, one isolate was fluoroquinolone resistant and one isolate combined both features. With the exception of C/V and AMK, which were not detected in other isolates, the multi-drug-resistant phenotype that affected 16 different substances was combined of partial multi-drug-resistance phenotypes found in other isolates. The phenomenon of multi-drug-resistance is described by Nikaido 2009 [20].

### 3.5. Prevalence of Antimicrobial Resistance Genes

Isolates differed also in their genotypic resistance. Calculated back to the initial number of meat samples, the prevalence of *E. coli*-borne antimicrobial resistance genes was as shown in Table 15.

### 3.6. Back-Tracking of Antimicrobial Resistant E. coli to Farm

We also tried to assess the chromosomal similarity between *E. coli* isolated from meat at slaughter and pig feces at the farm by amplicon typing. This was done for farms E5 and A6, where the highest number of isolates was collected. In general, isolates clustered repeatedly with fecal isolates from the farm, but not necessarily from the same animal: Going through the results of farm A6 in more detail (Figure 1), belly meat isolates of pig-p18 clustered with fecal isolates of pig p17. Belly meat isolate p20-2 clustered with belly meat isolate p11-1 and also with a fecal isolate of pig p11. Other belly meat isolates of pig p20 clustered with each other and with a fecal isolate of p20. In addition, that cluster included isolates of other pigs as well. Pig p11–p20 were all slaughtered in the same lot.

Having observed that, we wanted to assess whether the spread of similar isolates occurred before slaughter or rather afterwards in the form of cross-contamination. Therefore, we used isolates from different slaughter lots/fattening periods in the cluster analysis of farm E5, and included also isolates from the intestinal tract taken at slaughter (cecum/rectum; Figure 2). One belly meat isolate (p18) clustered with fecal isolates from the next fattening period and also with isolates from a sow housed in a different housing. Isolates taken from the intestinal tract at slaughter clustered with fecal isolates from the same pig at prior samplings (p31) and with pigs of the same slaughter lot, but also with pigs of prior/later lots from the same farm, and even with isolates from the farmer, who was not involved in slaughtering (Figure 1).

Exemplarily, we tried to trace back the antimicrobial resistance of isolates back to the farm. We found several isolates that matched in their ERIC-amplicon patterns and AMR-phenotypes, but most of them differed in genotypes (Table 16). On a farm level—pairing meat isolates not only within the same animal, but also with other animals from the same farm—slightly more similarities in genotypes were seen (e.g., E5, S10—control with S43—meat).

## 4. Discussion

Contamination rates of meat with *E. coli* were between 5.6 and 11.1% at the slaughterhouses of organically produced pigs, while between 16.7% and 36.8% of the samples were positive at the slaughterhouses where conventionally produced pigs were slaughtered. That difference might be related to differences in the slaughter procedure, which, however, could not be substantiated due to the lack of detailed information. When comparing results to a study of Schwaiger et al. [10], even higher prevalences of 72% were found.

*Escherichia coli* isolates at slaughter were phenotypically resistant—or had MIC-values above the ECOFF—against/for up to 19 different substances (out of 26 tested). Resistance or reduced AMS affected, at least in single cases, all substances except meropenem. Although the spectrum of tested substances was not completely comparable, the core resistance pattern of porcine *E. coli* in this study—AMP-AMC-PIP-DOX—was comparable to the core pattern found by Schwaiger et al. [10].

No obvious correlation could be detected between antimicrobial use and resistance. To give one example, beta-lactam resistance was more pronounced in A21, which did not use beta-lactams, compared to A6, which did. However, we saw a certain correlation with overall 1-year treatment frequency, which was assessed in another part of the study (data not shown). Overall, resistance was most pronounced in isolates from A21 and the fattening sector of A21 had a 1-year treatment frequency of 6.38, which was above the third quartile of 50 farms from South Germany (6.35; median 1.0). Compared to that, the fattening sector of M13 had a 1-year treatment frequency of 1.12 (0.62 in suckling piglets, 4.99 in sows), and E5 had a 1-year treatment frequency of only 0.06 in fattening pigs, but 1.69 in suckling piglets and 8.7 in sows. H18 and J20—the two organic farms—had a 1-year treatment frequency of 0.0 and 0.04, respectively. Unfortunately, no treatment frequency could be assessed for farm A6, since the farm left that part of the study prematurely. Treatment frequency was repeatedly described as an important enhancer of selection (e.g., Cobey et al., 2017 [21]) and is the subject of continuous monitoring in German fattening pigs now (Flor et al., 2022 [22]).

The spectrum of clinical resistances in isolates at slaughter was much broader in farm E5, compared to farm A6. Farm E5 combined several practices that are known to favor the selection of antimicrobial resistance: First, and as a routine, piglets were treated very early in life, with the very first additional feeding offered when they still fed on milk. Early exposure to antimicrobials had been shown to be linked with delayed maturation of microbiomes and lowered alpha-diversity [23], which depicts selection. Second, a broad-spectrum antibiotic (chlortetracycline) was applied, while broad spectrum antibiotics are known to aggravate selection [24]. Third, two antibiotics were combined when treating the piglets, which is also known to increase selection [25], at least when done without appropriate diagnostics, and apart from antibiotics which exploit synergistic pathways like sulfonamides plus trimethoprim. Fourth, a broad variety of different antibiotics was used in the farm (14 substances from 9 classes)—a practice also proven to select AMR [26]. Anyhow, unfavorable treatments were carried out in farm A6 as well, where pigs of one lot were treated consecutively with four different antibiotics during a total time of 23 days.

In farm E5, all fattening pigs were bred on the farm. This does not necessarily favor antimicrobial resistance—on the contrary, purchasing piglets from elsewhere is suspected to negatively affect the health state of pigs and thus to increase the need for treatment. However, it helps resistant isolates, once selected, to permanently establish, especially if antibiotic selection is continued as a prophylactic routine practice. To be clear, routine treatment is clearly discouraged by German veterinary guidelines (https://www.bundestieraerztekammer.de/btk/downloads/antibiotika/AB_Leitlinien2015_EN.pdf; accessed on 19 July 2022). Anyhow, such routine treatment is kind of “accepted reality”—albeit with gritted teeth: Guideline 3 states that the use of antimicrobials always requires a (clinical plus minus laboratory) diagnosis, but gives further advice in case that antibiotics are used “at regular intervals for repeated or long-term use in animal groups or herds”. In that case, susceptibility testing is required. The seeming discrepancy between both passages (obligatory diagnosis/application at “regular intervals”) is best explained by cases like farm E5: Indeed, a clinical diagnosis was documented for all piglets, so this prerequisite did not hinder the farm to apply routine treatment. In such cases, the specified guideline on repeated treatment ensures susceptibility testing, at least (and might put off farmers from applying routine treatment due to diagnostic efforts). In 2018, guidelines had been made legally binding by incorporating them into national legislation (“Tierärztliche Haus-apothekenverordnung (TÄHAV)”).

One point to mention is that we used phenotypic resistance only as a marker for phenotypic similarity between isolates, not as a therapeutic forecast. Therefore, we included substances for which no clinical breakpoints exist, and our results should not be misread as clinical diagnostics. In cases where breakpoints were available (all but 3), we used EUCAST-breakpoints, which tend to be set more cautious than CLSI-breakpoints [27] and were not specified for livestock. This approach was chosen since resistance of isolates at slaughter might, in the event of transfer, be relevant for human therapy, not for veterinary therapy.

For the same reason, we included substances that are not approved for livestock, but might be affected by co- and crossresistance, as proven before [10].

Looking at the zoonotic relevance of our general findings, focus is surely aimed at resistance to critically important antimicrobials. Third and fourth-generation cefalosporines as well as quinolones were affected by clinical resistance. Up to 13% (in total: 6.3%) of meat samples were contaminated with *E. coli* that carried *bla*_CTX-M_. Comparing German data of 2017, the prevalence of presumptive ESBL-isolates in fattening pigs was 41.8%, while it was 4.9% in pork meat. These data are in the range of our findings for meat, and in the upper range within Northern and Western Europe [28]. Since ESBL-*E. coli* might colonize the human intestine, these bacteria might act as reservoirs or opportunistic pathogens at any time later in life, as illustrated by Hölzel et al. [29]. However, Sharp et al. [30] do not see major indication for meat consumption playing a role in ESBL-colonization, and vegetarians bear more risk of being colonized by antimicrobial-resistant bacteria then humans with an omnivore life style [31]. The latter might be connected to biasing factors in the life style of vegetarians, such as having stayed abroad [32].

As limitations of our study, it has to be mentioned that we could not synchronize the different analyses, so that the isolates that underwent phenotyping and genotyping were not analyzed by amplicon typing, and isolates that were amplicon typed were not phenotyped or genotyped, vice versa. However, when we did so in another part of the study with isolates from humans and pigs that also resembled each other in ERIC-patterns, we found that they did not resemble each other in phenotypes, genotypes, and plasmid typing (data not shown). This is no surprise, since AMR features in Enterobacteriaceae are thought to be mainly mobile [33,34] while isolates themselves, of course, might be clonally spread—as it is very well known for pathogenic serovars of *E. coli* [35].

Genotypic and phenotypic profiles were not congruent in most cases, but rather two different features of the same organism. One exception of that rule was doxycycline “resistance” (means: MIC above ECOFF), where the most common causative genes in porcine *E. coli*—*tet*(A) and *tet*(B) [36]– were included. In that case, we could prove good accordance between genotypes and phenotypes: All doxycycline resistant isolates that were genotyped had at least one of the two causative genes. When matching genotypes with genotypes in six pairings between isolates from feces and pork, *tet*(A) was shared in five of the six pairings where it was present, while *tet(*B) was not shared between isolates of both pairs where it had occurred in one of the isolates. Given the high prevalence of *tet*(A) in porcine *E. coli*—Schwaiger et al. [36] found a prevalence of 57.7% in the same geographical region—the simultaneous occurrence of *tet*(A) in two isolates has limited significance as an epidemiologic link. Of other genes, *str*A-*str*B occurred in six pairings, but were shared in only two. One pair shared all four resistance genes that were detected, namely *tet*(A), *str*A-*str*B, and *sul*2. However, of seven detected phenotypic markers, only three were shared within that pair.

Looking at the amplicon-based cluster analysis, we saw similarities (i) between isolates of different lots at farm, (ii) between isolates from pigs on farm and at slaughter, and (iii) between isolates from pigs at slaughter and from the farmer. The GelCompare Software was allowed to include weak bands, so that the human eye would see less similarity than the software. In addition, we chose a high position tolerance for “identity” between bands, due to a low reproducibility of patterns: one strain, included as positive control in each gel, generated different band patterns with identity values as low as 63.4%, when no position tolerance was applied. However, even the visual judgment resulted in high similarity between one isolate at slaughter and two fecal isolates from the farmer, sampled months ago (Figure 1—lower part). Since the farmer was not present at slaughter, any exchange or colonization from a third, common source must have occurred on the farm, not at slaughter. One should take into account that the discriminative power of amplicon-typing is limited (Wilson & Sharp, 2006) and no whole genome sequencing was performed yet. Anyhow, the analysis revealed plausible results by identifying clones within the four fecal isolates monthly taken from the same pig. Furthermore, MLST-analysis (https://enterobase.warwick.ac.uk/species/ecoli/allele_st_search, accessed on 19 July 2022)—exemplarily performed in other cases where isolates form pigs and farmers resembled each other in ERIC profiles—found up to 99.9% identity of base pairs between ERIC-clones (5351 of 5356 bp), although other ERIC-clones had lower similarity in the MLST analysis (minimum 99.2% or 5407 of 5451 bp). The fact that Marshall et al. [37] could prove the spread of labelled *E. coli* from the intestine of cattle and pigs to farmers already in 1990 renders our observations plausible.

The spread of bacteria does not necessarily mean spread of antimicrobial resistance, as illustrated by the low congruence of antimicrobial resistance in feces-pork-pairs, as well as the high variability of phenotypic resistance profiles in 29 *E. coli* isolated from the same piece of meat. To look into those 29 isolates gives the impression of seeing recombination at work, since 14 of 16 antimicrobial resistances found in the most multiresistant isolate were also found in other isolates—in different combinations and with different frequency. No mating experiments could be performed with multiresistant strains, since no S2-facilities were available at the time and place of the study. However, we do not expect further insights from an in-vitro experiment, compared to the in-vivo indication that recombination probably had happened. All phenotypic resistances that formed part of the multiresistance profile had already been proven by others to be transferrable [38,39,40,41,42,43,44,45]. In total, 8.1% of all meat isolates were positive for *sul*I plus *qac*EΔ1, indicating the presence of integrons (data not shown), which confirms the reports of others [46]. Thus, once meat is contaminated with antimicrobial-resistant pathogens, this might set humans at risk—at least the risk of introducing transferrable resistance genes into their intestines [47,48]. However, simple measures of kitchen hygiene, such as careful handling and cooking, might prevent that risk [8].

## 5. Conclusions

Finally, *E. coli* isolated from meat samples occasionally showed resistance or reduced susceptibility to almost all of the substances investigated. As a limitation of our study, we state that we could only do the whole set of investigations in a selected subgroup of isolates. We characterized all isolates from meat, but finding the source of their (partly mobile) antimicrobial resistance features within thousands of isolates from the farm resembles the famous search for the needle in a haystack. Selective approaches to isolate resistant bacteria and plasmid characterization might help to narrow that search in future studies. Anyhow, looking at the relationship between isolates at a fingerprint-level, we have strong indications that antibiotic-resistant bacteria are transferred from farms to meat at slaughter. That means that these isolates can be further transferred to humans through food, although this risk can be minimized by simple measures of kitchen hygiene. In total, our results point towards the fact that fecal (cross-)contamination is, without any doubt, an important source of meat contamination with potential pathogens, and that antimicrobial (mis)use on farms selects for antimicrobial-resistant *E. coli.* However, the contamination of meat with antimicrobial-resistant agents is hard to predict from fecal isolates, since the (cross-)contamination of meat is unpredictable and the carriage of antimicrobial resistance genes is a strikingly dynamic process.

## Figures and Tables

**Figure 1 microorganisms-10-01485-f001:**
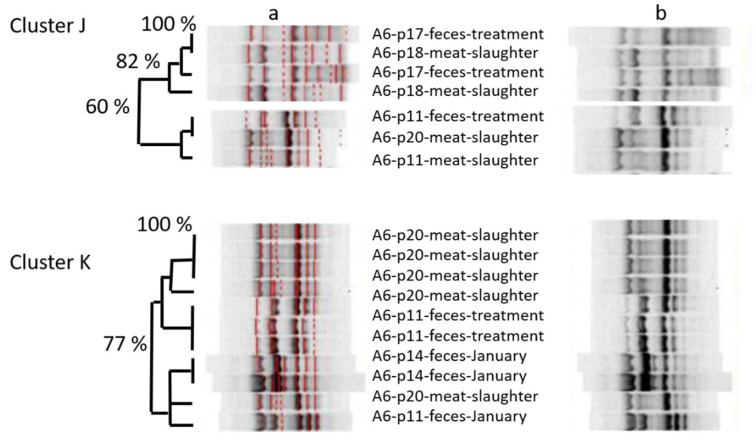
ERIC-amplicon-based cluster analysis of isolates from farm A6, taken at farm and at slaughter. Detail from the analysis of 2766 isolates in total, thereof 657 from farm A6. DNA-fragments were amplified by ERIC primers and visualized on 1% Agarose gel. All gels were aligned to standard bands: (**a**) software-based similarity assessment; (**b**) aligned, but unmarked bands for visual judgement. Numbers p11, p17, p18, p14, and p20: pigs tracked from November to January. All pigs slaughtered at the end of the tracking period. Cluster J, K: separate clusters with less than 50% similarity.

**Figure 2 microorganisms-10-01485-f002:**
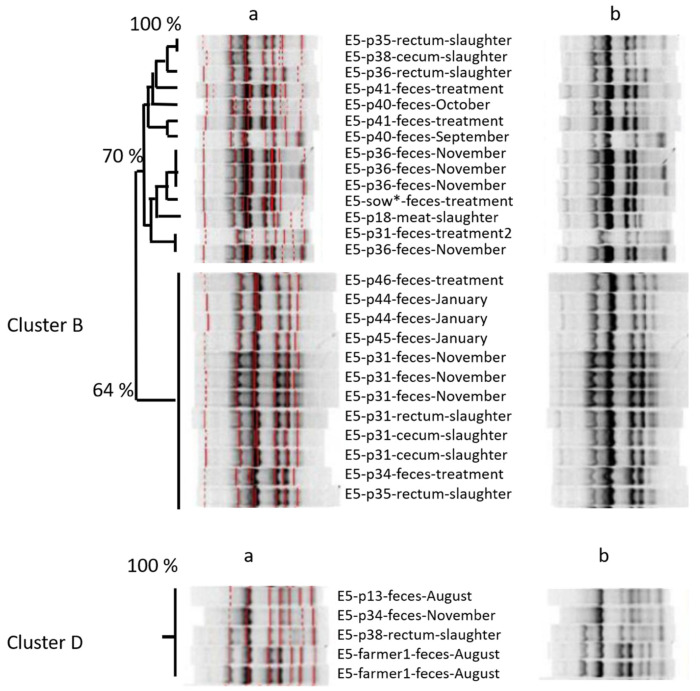
ERIC-amplicon-based cluster analysis of isolates from farm E5, taken at farm and at slaughter. Detail from the analysis showed 2766 isolates in total, with 1195 from farm E5. DNA-fragments were amplified by ERIC primers and visualized on 1% Agarose gel. All gels were aligned to standard bands: (**a**) software-based similarity assessment; (**b**) aligned, but unmarked bands for visual judgement. Numbers p13, p18: pigs tracked from May to August. Numbers p31, p34, p35, p36, p38, and p40: pigs tracked from August to November. Numbers p41, p44, p45, and p46: pig tracked from January to April. All treatments during the first month of tracking. All pigs slaughtered at the end of their tracking period. Sow: mother of p47 (pig fattened together with p41, p45, p44, and p46) during the first treatment days of p47. Cluster B, D: separate clusters with less than 50% similarity.

**Table 1 microorganisms-10-01485-t001:** Characteristics of included farms.

Farm	Husbandry Type	Herd Size (*n*)	Production Type	Occupancy	Fattening Period (Days)
A6	conventional	500	fattening	all in/all out	approx. 109
A21	conventional	960	fattening	all in/all out	approx. 115
E5	conventional	1333	piglet production, fattening	all in/all out	approx. 100
H18	organic	41	fattening	continuous	approx. 126
J20	organic	80	fattening	all in/all out	approx. 135
M13	conventional	978	piglet production, fattening	continuous	approx. 100

**Table 2 microorganisms-10-01485-t002:** Overview of antimicrobial agents used for phenotypic resistance testing.

Antibiotic Class	Antibiotic	Abbreviation	MIC Range	Breakpoints (mg/L)		Source
				S (≤)	R (>)	
ß-Lactams						
Penicillins	Amoxicillin/Clavulanate	AMC	1/2–64/8	8/2	8/2	EUCAST
	Ampicillin	AMP	1–128	8	8	EUCAST
	Piperacillin	PIP	1–128	8	16 ^a^	EUCAST **
	Piperacillin Tazobactam	PIT	1/4–128/4	8/4	16 ^a^/4	EUCAST **
Cephalosporins	Cefepim	CEP	0.25–32	1	4	EUCAST
	Cefotaxime	CTX	0.25–32	1	2	EUCAST
	Cefoxitin	COX	2–16	8	8	EUCAST
	Cefpodoxim/Clavulanate	C/V	0.25/4–32/4	1/4	1/4	EUCAST
	Cefpodoxim-Proxetil	CPP	0.25–32	1	1	EUCAST
	Ceftazidim	CAZ	0.25–32	1	4	EUCAST
	Cefuroxim	CXM	2–16	8	8	EUCAST
Carbapeneme	Ertapenem	ERT	0.25–2	0.5	1 ^b^	EUCAST **
	Imipenem	IMP	0.25–32	2	8 ^c^	EUCAST **
	Meropenem	MER	0.25–32	2	8	EUCAST
Aminoglycosides	Amikacin	AMK	4–32	8	16 ^a^	EUCAST **
	Gentamicin	GEN	0.25–32	2	4 ^d^	EUCAST **
	Tobramycin	TOB	0.25–32	2	4 ^d^	EUCAST **
Fenicole	Florfenicol	FLL	0.5–64		16*	ECOFF *
Fluorchinolone	Ciprofloxacin	CIP	0.0625–8	0.5 ^e^	1 ^b^	EUCAST **
	Enrofloxacin	ENR	0.0625–8		0.125 *	ECOFF *
	Levofloxavin	LEV	0.0625–8	1 ^e^	2 ^f^	EUCAST **
Tetracycline	Doxycylin	DOX	0.125–16		4 *	ECOFF *
Others	Fosfomycin	FOS	16–128	32	32	EUCAST
	Trimethoprim/Sulfamethoxazol	T/S	0.5/9.5–4/76	2	4	EUCAST
	Aztreonam	AZT	1–16	1	4	EUCAST
	Colistin	COL	1–8	2	2	EUCAST

* ECOFF, not indicating clinical resistance, but reduced susceptibility to the respective antibiotic. Source: EUCAST. ** former version, current value as indicated below: ^a^ Current EUCAST-breakpoint table: 8 (1 January 2022); ^b^ Current EUCAST-breakpoint table: 0.5 (1 January 2022); ^c^ Current EUCAST-breakpoint table: 4 (1 January 2022); ^d^ Current EUCAST-breakpoint table: 2 (1 January 2022); ^e^ Current EUCAST-breakpoint table: 0.25 (1 January 2022); ^f^ Current EUCAST-breakpoint table: 1 (1 January 2022).

**Table 3 microorganisms-10-01485-t003:** Target genes, fragment size, and annealing temperature of primers used for qualitative resistance gene detection.

Number *	Primer FW/RV	Target Gene	Sequence (Direction 5′-3′)	Fragment Size (bp)	Annealing Temperature (°C)	Source for Primers
2	AMPC FW	chromosomal encoded *amp*C	GATCGTTCTGCCGCTGTG	271	57	Corvec et al., 2007 [13]
	AMPC RV		CCGTCAACTTTCGCGTATTT			
3	*bla*_CTX-M_U FW	conserved *bla*_CTX-M_-region	ATGTGCAGYACCAGTAARGT	593	50	Pagani et al., 2003 [14]
	*bla*_CTX-M_U RV		TGGGTRAARTARGTSACCAGA			
12	*qac*E∆1 FW	qacE∆1	GGCTTTACTAAGCTTGCCCC	203	57	Bischoff et al., 2012 [15]
	*qac*E∆1 RV		AGCCCCATACCTACAAAGCC			
14	*str*(A) FW	*str*(A)	CCTGGTGATAACGGCAATTC	546	53	Lanz et al., 2003 [16]
	*str*(A) RV		CCAATCGCAGATAGAAGGC			
15	*str*(B) FW	*str*(B)	ATCGTCAAGGGATTGAAACC	509	54	Lanz et al., 2003 [16]
	*str*(B) RV		GGATCGTAGAACATATTGGC			
16	*sul*(I) FW	*sul*(I)	TTCGGCATTCTGAATCTCAC	822	53	Maynard et al., 2003 [17]
	*sul*(I) RW		ATGATCTAACCCTCGGTCTC			
17	*sul*(II) FW	*sul*(II)	CGGCATCGTCAACATAACC	722	59	Maynard et al., 2003 [17]
	*sul*(II) RW		GTGTGCGGATGAAGTCAG			
18	*tet*(A) FW	*tet*(A)	GTGAAACCCAACATACCCC	888	55	Maynard et al., 2003 [17]
	*tet*(A) RW		GAAGGCAAGCAGGATGTAG			
19	*tet*(B) FW	*tet*(B)	TACGTGAATTTATTGCTTCGG	206	60	Aminov et al., 2002 [18]
	*tet*(B) RW		ATACAGCATCCAAAGCGCAC			

***** According to Appendix A.

**Table 4 microorganisms-10-01485-t004:** PCR amplification conditions.

Program	Cycles	Target Temperature (°C)	Hold Time (s)
Pre-incubation	1	94	300
Amplification	35	94	60
		*	**
		72	***
Final Extension	1	72	300
Cooling	1	4	/

* see Table 3. ** primer pair number 2, 12: 30 s; number 3: 40 s; number 14–19: 60 s. *** primer pair number 2, 19: 30 s; number 3, 12: 60 s; number 14–18: 90 s.

**Table 5 microorganisms-10-01485-t005:** ERIC-PCR protocol.

Program	Cycles	Target Temperature (C°)	Hold Time (s)
Pre-incubation	1	95	420
Amplification	30	90	30
		52	60
		65	480
Final Extension	1	65	960
Cooling	1	4	/

**Table 6 microorganisms-10-01485-t006:** Target gene, fragment size, and annealing temperature of the primers used for ERIC-PCR.

Number *	Primer FW/RV	Target Gene	Sequence (Direction 5′-3′)	Fragment Size (bp)	Annealing Temperature (°C)	Source for Primers
5	ERIC FW ERIC RV	ERIC	ATGTAAGCTCCTGGGGATTCAC AAGTAAGTGACTGGGGTGAGCG	variable	52	Versalovic et al., 1991 [19]

***** According to Appendix A.

**Table 7 microorganisms-10-01485-t007:** Antibiotics applied in study animals.

Antibiotic	Farm	Pigs (n/Study N)	Treatment Days
Amoxicillin-trihydrate	A6	20/30	2–6
Colistin-sulfate	A6	20/30	3–10
	A21	10/20	7
	E5	40/40	7 *
	M13	10/20	5
Oxytetracycline	E5	40/40	1 **
Sulfadiazin + trimethoprim	M13	10/20	9
Tetracycline-hydrochlorid	E5	40/40	7 ***
Tylosin-tartrate	A6	30/30	3–6

* combined with tetracycline-hydrochlorid; ** applied on the first day of life; *** combined with colistin.

**Table 8 microorganisms-10-01485-t008:** Antibiotics applied at the respective farms during the study period: overview of all animals.

Farm	Production Mode	Antibiotics Used during Study Period ^#^
A6	conventional	**Amoxicillin-trihydrate, Colistin-sulfate**, Florfenicol, **Tylosin-tartrate**
A21	conventional	**Colistin-sulfate**, Tylosin-phosphate
E5	conventional	Amoxicillin-trihydrate, **Colistin-sulfate**, Enrofloxacin, Gentamicin-sulfate, **Oxytetracycline**, Sulfadiazin + Trimethoprim, **Tetracycline-hydrochlorid**, Tylosin
H18	organic	no antimicrobials
J20	organic	Cefquinome-sulfate, Chlortetracycline-hydrochlorid, Sulfathiazol + Sulfadimidin + Trimethoprim
M13	conventional	**Sulfadiazin + Trimethoprim**, Amoxicillin-trihydrate, Benzylpencillin-benzathin, Benzylpenicillin-procain, Cefquinome-sulfate, Chlortetracycline-hydrochlorid, Dihydrostreptomycin-sulfate, Enrofloxacin, Lincomycinhydrochlorid-monohydrate, Spectinomycin-sulfate-tetrahydrate, Sulfadoxin, Tildipirosin

# Only antibiotics in **bold** were used in study animals, others were applied in other pens or age groups.

**Table 9 microorganisms-10-01485-t009:** Occurrence of *E. coli* on meat, farmwise description.

Farm	Mode of Farming	n Positive per n Animals at Slaughter (%)	n Positive per n Meat Samples (%)	Number of Isolates
A6	conventional	16/30 (53.3%)	18/60 (30.0%)	53
A21	conventional	10/19 (52.6%)	14/38 (36.8%)	66
E5	conventional	14/37 (37.8%)	16/74 (21–6%)	49
H18	organic	1/9 (11.1%)	1/18 (5.6%)	4
J20	organic	1/9 (11.1%)	2/18 (11.1%)	2
M13	conventional	3/18 (16.7%)	3/18 (16.7%)	24

**Table 10 microorganisms-10-01485-t010:** Phenotypic susceptibility of meat-borne *E. coli* against beta-lactams: penicillins and carbapenems.

Farm	Beta-Lactams Used	*E. coli* (*n*)	MIC50/MIC90 *						
			AMC	AMP	PIP	PIT	ERT	IMP	MER
A6	yes ^#^	53	16/64	256/256	128/256	n.d.	n.d.	n.d.	n.d.
A21	no	66	64/128	256/256	256/256	n.d.	0.25/0.25	n.d.	n.d.
E5	yes ^$^	49	64/128	256/256	256/256	2/4	n.d.	0.25/16	n.d.
H18	no	4	4/16	2/256	2/256	n.d.	n.d.	n.d.	n.d.
J20	yes ^$^	2	n.d. *	4/16	n.d.	n.d.	n.d.	n.d.	n.d.
M13	no	24	2/32	2/256	1/128	1/2	n.d.	n.d.	n.d.

AMC = Amoxicillin-clavulanate, AMP = Ampicillin, PIP = Piperacillin, ERT = Ertapenem, IMP = Imipenem, MER = Meropenem. ^#^ in study animals ^$^ in other animals n.d.: not determined, MIC 100 below breakpoint/ECOFF. * In case that at least one isolate had a MIC-value above the EUCAST-breakpoint, the MIC50/MIC90-value is listed. Otherwise not determined (n.d., all isolates susceptible).

**Table 11 microorganisms-10-01485-t011:** Phenotypic susceptibility of meat-borne *E. coli* against beta-lactams: cefalosporines.

Farm	Beta-Lactams Used	*E. coli* (*n*)	MIC50/MIC90 *						
			CEP	CTX	COX	C/V	CPP	CAZ	CXM
A6	yes ^#^	53	n.d.	n.d.	n.d.	n.d.	n.d.	n.d.	n.d.
A21	no	66	0.25/0.25	0.25/0.25	8/32	0.5/1	0.5/0.5	0.25/1	4/8
E5	yes ^$^	49	0.25/0.25	0.25/0.25	8/32	0.5/1	0.5/1	0.25/0.5	4/8
H18	no	4	n.d.	n.d.	n.d.	n.d.	n.d.	n.d.	n.d.
J20	yes ^$^	2	0.25/8	0.25/16	8/32	0.5/8	0.5/8	0.25/8	8/32
M13	no	24	n.d.	n.d.	4/8	n.d.	n.d.	0.25/0.25	4/8

CEP = Cefepim, CTX = Cefotaxime, COX = Cefoxitin, C/V = Cefpodoxime-clavulanate, CPP = Cefpodoxime-proxetil, CAZ = Ceftazidim, CXM = Cefuroxime. ^#^ in study animals ^$^ in other animals n.d.: not determined, MIC 100 below breakpoint/ECOFF. * In case at least one isolate had a MIC-value above the EUCAST-breakpoint, the MIC50/MIC90-value is listed. Otherwise, not determined (n.d., all isolates susceptible).

**Table 12 microorganisms-10-01485-t012:** Phenotypic susceptibility of meat-borne *E. coli* against-aminoglycosides (AG) and fluoroquinolones (FQ).

Farm	AG or FQ Used	*E. coli* (*n*)	MIC50/MIC90 *					
			AMK	GEN	TOB	CIP	ENR **	LEV
A6	no	53	n.d.	n.d.	n.d.	n.d.	0.063/0.063	n.d.
A21	no	66	4/8	n.d.	0.5/1	0.063/1	0.063/1	0.063/1
E5	both ^$^	49	n.d.	1/2	n.d.	0.063/0.5	0.063/1	0.063/1
H18	no	4	n.d.	n.d.	n.d.	n.d.	0.063/0.063	n.d.
J20	no	2	n.d.	n.d.	n.d.	n.d.	0.063/0.063	n.d.
M13	no	24	n.d.	n.d.	n.d.	n.d.	0.063/0.063	n.d.

AMK = Amikacin, GEN = Gentamicin, TOB = Tobramycin, CIP = Ciprofloxacin, ENR = Enrofloxacin, LEV = Levofloxacin. ^$^ in other animals n.d.: not determined, MIC 100 below breakpoint/ECOFF. * In case at least one isolate had a MIC-value above the EUCAST-breakpoint, the MIC50/MIC90-value is listed. Otherwise not determined (n.d., all isolates susceptible). ** all MIC50/MIC90 values listed since no EUCAST-breakpoints are available.

**Table 13 microorganisms-10-01485-t013:** Phenotypic susceptibility of meat-borne *E. coli* against diverse antibiotics.

Farm	Use of	*E. coli* (*n*)	MIC50/MIC90 *					
			FLL **	DOX **	FOS	T/S	AZT	COL
A6	FLL ^$^, COL ^#^	53	8/8	2/32	16/16	0.5/8	1/1	1/1
A21	COL ^#^	66	8/16	4/32	16/32	0.5/8	1/1	1/1
E5	COL ^#^, TET ^#^, SUL ^$^	49	8/16	8/32	16/64	0.5/8	1/1	1/1
H18	none	4	8/16	4/32	n.d.	0.5/8	n.d.	n.d.
J20	TET ^$^, SUL ^$^	2	8/128	2/4	16/128	n.d.	1/32	n.d.
M13	COL ^#^, SUL ^#^	24	8/16	4/16	16/16	0.5/0.5	n.d.	n.d.

FLL = Florfenicol, DOX = Doxycycline, FOS = Fosfomycin, T/S = Trimethoprim-sulfamethoxazol, AZT = Aztreonam, COL = Colistin. ^#^ in study animals ^$^ in other animals n.d.: not determined, MIC 100 below breakpoint/ECOFF. * In case at least one isolate had a MIC-value above the EUCAST-breakpoint, the MIC50/MIC90-value is listed. Otherwise, not determined (n.d., all isolates susceptible). ** all MIC50/MIC90 values listed since no EUCAST-breakpoints are available.

**Table 14 microorganisms-10-01485-t014:** Resistance Patterns of all *E. coli* isolates (*n* = 29) from the same piece of belly meat of a pig from farm A21.

Pattern of Phenotypic Resistance/Insusceptibility	n Affected Antibiotics	n Affected Classes **	n Isolates with That Pattern
AMC	1	1	1
DOX *	1	1	2
AMC AMP PIP	3	1	8
AMC AMP PIP COX	4	2	4
AMC AMP PIP DOX *	4	2	6
AMC AMP PIP FOS	4	2	1
AMC AMP PIP CXM DOX *	5	3	1
AMC AMP PIP DOX * T/S	5	3	3
AMC AMP PIP CIP ENR * LEV	6	2	1
AMC AMP PIP CEP CTX COX CPP CXM DOX * T/S AZT	11	5	1
AMC AMP PIP CEP CTX C/V CPP CXM AMK FLL * CIP ENR * LEV DOX * T/S AZT	16	7	1

AMC = Amoxicillin-clavulanate, DOX = Doxycycline, AMP = Ampicillin; PIP = Piperacillin, COX = Cefoxitin, FOS = Fosfomycin, CXM = Cefuroxime, T/S = Trimethoprim-sulfamethoxazol, CIP = Ciprofloxacin, ENR = Enrofloxacin, LEV = Levofloxacin, CEP = Cefepim, CTX = Cefotaxim, CPP = Cefpodoxime-proxetil, AZT = Aztreonam, C/V = Cefpodoxime-clavulanate, AMK = Amikacin, FLL = Florfenicol. * above EUCAST-ECOFF, no breakpoint provided. ** penicillins, cefalosporines and carbapenems treated as different classes.

**Table 15 microorganisms-10-01485-t015:** Prevalence of *E. coli* that carried antimicrobial resistance genes in meat samples at slaughter (*n* meat samples = 244).

Farm	Number of Meat Samples	% of Meat Samples Positive for *E. coli* That Carry…							
		*bla* CTXM	*qac*EΔ1	*str*(A)	*str*(B)	*sul*I	*sul*II	*tet*(A)	*tet*(B)
A6	60	6.7	6.7	10.0	8.3	5.0	11.7	15.0	6.7
A21	38	13.2	7.9	18.4	18.4	10.5	15.8	15.8	10.5
E5	74	1.4	5.4	10.8	4.1	2.7	6.8	13.5	8.1
H18	18	5.6	5.6	5.6	<lod	5.6	5.6	<lod	5.6
J20	18	<lod	<lod	<lod	<lod	<lod	<lod	<lod	<lod
M13	36	5.6	5.6	2.8	2.8	2.8	<lod	2.8	<lod
**total**	**244**	**5.3**	**5.3**	**9.4**	**6.6**	**4.5**	**7.8**	**10.7**	**6.1**

Iod = limit of detection (3.1 × 10^1^ cfu *E. coli* per cm^2^).

**Table 16 microorganisms-10-01485-t016:** Genotypic resistance patterns of *E. coli* from meat and feces (paired by pig of origin), which resembled each other in phenotypic AMS patterns.

Farm	Pig	Source	Time between Samplings	Phenotypic Resistance Pattern	Genotypic Resistance Pattern **
E5	S10	Feces (control)	< 1 month	**AMC AMP PIP DOX ***	*bla*CTXM-***tet*(A)**-*sul*(I)-*qac*EΔ1
		Neck meat		**AMC AMP PIP DOX** * T/S	***tet*(A)**-*tet*(B)-*str*(A)-*str*(B)
	S43	Feces (during treatment)	6 months	**DOX ***	*tet*(B)-*str*(A)-*str*(B)
		Belly meat		**DOX ***	*bla*CTXM-*tet*(A)-*sul*(I)
	S47	Feces (during treatment)	6 months	**AMC AMP DOX *** T/S	***tet*(A)-*str*(A)-*str*(B)**-*sul*(II)
		belly meat		**AMC AMP DOX *** PIP FOS	*bla*CTXM-***tet*(A) *str*(A)-*str*(B)**
A6	S1	Feces (during treatment)	4 months	**AMP PIP DOX *** AMC FLL * COL	***tet*(A)-*str*(A)-*str*(B)-*sul*(II)**
		neck meat		**AMP PIP DOX** * T/S	***tet*(A)-*str*(A)-*str*(B)-*sul*(II)**
	S7	Feces (during treatment)	4 months	**AMC AMP PIP DOX ***	***tet*(A)**-*str*(A)-*str*(B)-*sul*(II)-*qac*EΔ1
		neck meat		**AMC AMP PIP DOX** * T/S	***tet*(A)**
	S8	Feces (during treatment)	4 months	**AMC AMP PIP DOX *****T/S** -GEN TOB COL	***tet*(A)**-*str*(A)-*str*(B)-*sul*(II)
		neck meat		**AMC AMP PIP DOX** * T/S	***tet*(A)**

AMC = Amoxicillin-clavulanate, AMP = Ampicillin; PIP = Piperacillin, DOX = Doxycycline, T/S = Trimethoprim-sulfamethoxazol, FOS = Fosfomycin, FLL = Florfenicol, COL = Colistin, GEN = Gentamicin, TOB = Tobramycin. * above the EUCAST-ECOFF, no clinical breakpoint provided. ** phenotypes not completely covered by genotyping. Bold: genes shared between isolates of the same on farm and at slaughter.

## Data Availability

Available from the authors upon resonable request.

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
