# Peer review of "Tracking Antimicrobial Resistant E. coli from Pigs on Farm to Pork at Slaughter"

_microorganisms, 2022, doi:10.3390/microorganisms10081485_

Round 1

Reviewer 1 Report

Thank you authors for interesting study and findings. Suggestions and comments below for consideration.

Line 63: Suggest to define/explain 'organic'.

Table 10xx: Was there any statistical comparison to see if there was any correlation between the use of antimicrobials and resistance against the respective antimicrobials?

Table 12: A21 was observed with relatively higher prevalence of E. coli carrying antimicrobial resistant genes. Any hypothesis on this observation? Was there any other preceding studies conducted in the same country/region, thought these studies could be in different farms, useful for brief comparison.

Line 128: Was there any description in Materials & Methods about the confirmation of ESBL production? 

Line 176: Was there any results/findings on MLST?

Line 225: Were the concentration data available for all samples and corresponding isolates? Would authors consider comparing the concentration (count) of E. coli by different types of samples (meat, caecum), by the antimicrobial profiles to see if the counts affected/was affected by different parts of samples and antimicrobial resistant profiles? 

Line 266: When comparing prevalence to another study (by Schwaiger et al.), was the method was similar to that of in this study i.e. at least non-enrichment/quantitative?

Line 405: "...only as a marker" Suggest to indicate market in what sense e.g. marker for antimicrobial resistance in relation to the use of antibiotics? hygiene marker? etc.

Line 407: One could not make sense of the "(lit###)". Was it a typo?

Line 496: "...that antimicrobial (mis)use on farms selects for antimicrobial resistant E. coli. - While agree with authors that this is a general view, suggest to include more statistical analysis and further description/discussion about findings shown in tables 10xx to support this conclusion. 

Author Response

We would like to thank both reviewers for their comments, which we incorporated in our revision in order to increase the quality of our submission. In the following text we give the overview of the changes in the article.

Specific Responses:

Response to Reviewer 1:

Comment: Line 63: Suggest to define/explain organic

Response: We added the legal definition for organic farms (line numbers 83-84).

Comment: table 10xx: Was there any statistical comparison to see if there was any correlation between the use of antimicrobials and resistance against the respective antimicrobials?

Response: Due to the very different number of isolates per farm and even per sample, we did not perform statistic analysis here. We could have included "farm" or "sample" as a random effect to a linear mixed model (we did so in the past with other data). However, checking of the descriptive data already allowed to see that there is NO correlation, thus we did not see a need to proof that absence of correlation with statistical methods. Unfortunately, the treatment frequency, for which we saw hints for correlation, was not assessed in all farms, so we did not have the possibility to perform meaningful statistics, here. We added one paragraph to the text: "No obvious correlation could be detected when comparing the applied antibiotics with those that were affected by antimicrobial resistance (Table 10a-d)"; line 265-266., and we discuss the association between treatment frequency and AMR now in line 396-408.

Comment: Table 12: A21 was observed with relatively higher prevalence of E. coli carrying antimicrobial resistant genes. Any hypothesis on this observation? Was there any other preceding studies conducted in the same country/region, thought these studies could be in different farms, useful for brief comparison.

Response: We now provide a hypothesis to explain that finding in the discussion (see above; line 396-408) All farms were located in more or less the same area of South Germany, and prevalences had been found to be similar between regions in that area in former, unpublished studies.

Comment: Line 128: Was there any description in Materials & Methods about the confirmation of ESBL production?

Response: Not sure what is meant here. All isolates from ChromAgar ESBL were confirmed in the microdilution assay (confirming ESBL-production) and MALDI-TOF (confirming species ID). The microdilution assay included indicator substances with and without clavulanate, which allow to distinguish between ESBL-enzymes and AmpC-type enzymes (see Table 2). However, we did not perform double disk diffusion synergy testing here.

Comment: Line 176: Was there any results/findings on MLST?

Response: You are right, MLST was only performed in other isolates (not included in this part of the study) and is therefore only included in the discussion, not in the results. We deleted that information and refer to the method (means: its reference) in the discussion only (line 504).

Comment: Line 225: Were the concentration data available for all samples and corresponding isolates? Would authors consider comparing the concentration (count) of E. coli by different types of samples (meat, caecum), by the antimicrobial profiles to see if the counts affected/was affected by different parts of samples and antimcrobial resistant profiles?

Response: No, unfortunately not - we assessed that exemplarily only. For feces, we did not record concentrations of E. coli and picked only four isolates per sample. For meat, we cannot connect informations on sample level except for the one example we´ve shown. Farmwise, the mean number of isolates per positive sample was 1 to 8 per positive sample, and no correlation was seen between mean concentration and resistance.

Comment: Line 266: When comparing prevalence to another study (Schwaiger et al.), was the method similar to that of this study i.e. at least non-enrichment/quantitative?

Response: Yes, the same methodology was used there.

Comment: Line 405: „…only as a marker“ Suggest to indicate market in what sense e.g. marker for antimicrobial resistance in relation to use of antibiotics? Hygiene marker? Etc.

Response: We changed that accordingly and specified „marker for phenotypic similarity between isolates“ (line 442-443).

Comment: Line 407: One could not make sense oft he „(lit###)“. Was it a typo?

Response: We apologize for this mistake and deleted it, thank you!

Comment: Line 496: „…that antimicrobial (mis)use on farms selects for antimicrobial resistant E. coli. – While agree with authors that this is a general view, suggest to include more statistical analysis and further description/discussion about findings shown in tables 10xx to support this conclusion.

Response: We hope that the reported findings on therapy frequency further strengthen our conclusion, which is, apart from that observation in A21, also based on the discussed observations in farm E5.

Reviewer 2 Report

Dear Authors,

The introduction did not introduce me enough to the subject. I feel unsatisfied. It would be nice to extend it, among others about what antibiotics are specifically administered to animals for which conditions.

I have one comment to the authors, why did they use MICs for antibiotics used in humans? Is VetCast not supplying sufficient data?

The work requires editing corrections: e.g. line 407 - no reference, line 146 - use of 2x letters "s", CFU should be written with a capital letter.

Materials and methods: in the text (not in the supplement) it is necessary to specify the sequences of the primers used in the PCR reaction. The PCR reaction conditions should also be supplemented (quantify all ions from the buffer and additives used in 1 reaction).

Best regards.

Author Response

We would like to thank both reviewers for their comments, which we incorporated in our revision in order to increase the quality of our submission. In the following text we give the overview of the changes in the article.

Specific Responses:

Response to Reviewer 2:

Comment: The introduction did not introduce me enough to the subject. I feel unsatisfied. It would be nice to extend it, among others about what antibiotics are specifically administered to animals for which conditions.

Response: We now briefly refer to the most used antibiotics in livestock in Germany (line 32-41). The conditions of use (therapeutic, but prophylactic use still occurs) are discussed in line 416-417 and 428-441. For the sake of clarity, we added "prophylactic" in line 428 (now "prophylactic routine praxis").

Comment: I have one comment to the authors, why did they use MICs for antibiotics used in humans? Is VetCast not supplying sufficient data?

Response: We explain the reason in line 452-454: The risk of antimicrobial resistant isolates at slaughter does not refer to livestock (pigs are dead then and won´t be treated), but to humans, who might be infected due to consumption. This is the reason why we want to assess AMR which is relevant for humans. For that, we had to use breakpoints for humans, not for livestock. If slaughterhouse data were meant to give feedback to the farmers (for future treatments), this would be a good reason to apply veterinary specific breakpoints in other studies. However, this was not our study goal here.

Comment: The work requires editing corrections: e.g. line 407 – no reference, line 146 – use of 2x letters “s”, CFU should be written with a capital letter.

Response: We apologize for the mistakes. Line 170 plus 445: corrected CFU: we would prefer to leave that to the editor, there are different spellings of cfu in the literature, also in JAM, and mostly it is cfu there.

Comment: Materials and Methods: in the text (not in the supplement) it is necessary to specify the sequences of the primers used in the PCR reaction. The PCR reaction condition should also be supplemented (quantify all ions from the buffer and additives used in 1 reaction).

Response: Thanks for the note, we have added the composition of the master mix (line 184-187). Furthermore we have added the sequences and sources of the primers in Table 3 and Table 6.

Reviewer 3 Report

The manuscript by Bassitta et al. is well written and original, with important contribution to the understanding of the complex antimicrobial resistance puzzle of pig origin E. coli on farm to slaughter.

Line 13: within the abstract section the authors must define the study aim

Line 14: „we performed” – please avoid the using of personal verb forms. Please carefully revise this concern throughout the manuscript;

Line 17: please use “antimicrobials” instead of “antibiotics” throughout the manuscript; “critically” – please remove this term from the sentence;

Line 19: “22.1 %” – when you express overall prevalence values, please insert in brackets the values of 95% Confidence Interval;

Line 20: “ECOFF” please explain this acronym;

Line 23: “16fold-multiresistance” – space insertion

Line 31: within the introduction section, a clear definition of pathogenic E. coli is needed, mentioning the classification of pathotypes, in order to highlight the study importance. Also, from public health point of view, the authors must highlight the importance of monitoring of E. coli within the food-chain, also in other animal origin foodstuffs (e.g. cheese - doi: 10.3390/antibiotics11060721 or beef - doi: 10.3390/foods9111543). This articles can be consulted and cited in order to increase the MDPI journals.

Line 129: please provide a reference for EUCAST

Line 139: please replace „sensitive” with „susceptible”

Table 2: in the first column, please avoid the unnecesary bolding of antimicrobial classes (e.g. aminoglycosides, fenicole ...)

Line 127: please provide the selection strategy of the used antimicrobials

I recommend the inclusion of Tables 4 and 5 in supplementary material files

Line 226: „^” – unclear symbol

Line 283: „multi-drug resistance” instead of „multiresistance” . Please define this phenomenon in insert the most representative reference for this (e.g. doi: 10.1111/j.1469-0691.2011.03570.x)

Lines 492-493: „Consumption of raw meat contaminated with (resistant) E. coli presents a health risk to the consumer” – this cannot be considered a conclusion of this work

Within the conclusion section the authors must emphasizes the study limitation and further perspectives

Reference list: please carefully italicize the scientific name of species throughout the references (e.g. line 528 – for Enterococcus faecium; and 530-531 for Escherichia coli, etc.)

Author Response

We would like to thank both reviewers for their comments, which we incorporated in our revision in order to increase the quality of our submission. In the following text we give the overview of the changes in the article.

Specific Responses:

Response to Reviewer 3:

Comment: Line 13: within the abstract section the authors must define the study aim

Response:  You are right, we only described the knowledge gap, which we aimed to fill, but did not link it to our study. We changed the wording of the third sentence, in order to shape a link to our study aim. We also ask for indulgence that we could not add another sentence to the abstract, due to word restrictions. Instead, we added a more explicit definition of the study aim in line 74-75 of the introduction.

Comment: Line 14: „we performed” – please avoid the using of personal verb forms. Please carefully revise this concern throughout the manuscript

Response: To our knowledge, there is no obligatory rule to avoid personal verb forms in scientific studies. Yet, there are language guidelines discouraging the use of passive voice. Thus, in our opinion this is merely a question of style, so we would prefer not to change it.

Comment: Line 17: please use “antimicrobials” instead of “antibiotics” throughout the manuscript; “critically” – please remove this term from the sentence

Response: We agree that antimicrobials instead of antibiotics would be more consistent with AMR, and fits better with the classical definition where antibiotics had been separated form chemotherapeutics. However, there is also a need for optimized machine search conditions nowadays, and especially the term “antibiotic use” is much more frequently used than is “antimicrobial use” (10fold). So, we changed “antibiotic” to “antimicrobial” whenever it is linked to resistance, but we kept “antibiotics” at other instances (and also in the headings of tables, for sake of shortness).

The expression “critically important antimicrobials” is a defined phrase coined by the WHO, so we kept “critically”.  

Comment: Line 19: “22.1 %” – when you express overall prevalence values, please insert in brackets the values of 95% Confidence Interval

Response: There is no confidence interval for prevalences here, since these values are not based on statistical assessement – they do not refer to the whole population, but only to the tested lot. One could calculate confidence intervals based on method accurracy; however, I have never seen this in other studies in the field, and validation (which we performed) is normally based on MIC-values, not on resistance prevalence. Anyhow, since we assured that there were no major (i.e. category changing) errors with the reference strain tested together with each lot, we can conclude that accuracy was 100 %, so, again, we would not have to state a confidence interval. We specify the repeated testing of a reference strain in the methods section now.

Comment: Line 20: “ECOFF” please explain this acronym

Response: Thank you for pointing this out, the definition has been added in the main text (line 162). In the abstract, we are very limited in words and its one quick search in the web which brings the reader to the definition (which, in addition, should be a known term for the addressed readership). In the abstract, we connected ECOFF to breakpoint by use of a slash, in order to give a hint that this is something similar.

Comment: Line 23: “16fold-multiresistance” – space insertion

Response: We have added a space, thank you!

Comment: Line 31: within the introduction section, a clear definition of pathogenic E. coli is needed, mentioning the classification of pathotypes, in order to highlight the study importance. Also, from public health point of view, the authors must highlight the importance of monitoring of E. coli within the food-chain, also in other animal origin foodstuffs (e.g. cheese - doi: 10.3390/antibiotics11060721 or beef - doi: 10.3390/foods9111543). This articles can be consulted and cited in order to increase the MDPI journals

Response: Thank you for the opportunity to clarify that. In E. coli (as in other zoonotic agents), antimicrobial resistance is often not linked to virulence, and antibotic treatment is often even contraindicated in pathogenic serovars (like EHEC). The risk linked to zoonotic transfer of AMR-E.coli is that of a reservoir organism introduced into the gut, which might there transfer its genes to other, more pathogenic bacteria, or which might occasionally develop into an opportunistic pathogen, e.g. infecting the bladder. This is very frequently mistaken, so it is our fault to not have addressed that more explicitly in the introduction. We added a passage dealing with that topic now (line 54-59).

We are aware of the fact that reviewers are requested to suggest MDPI-citations. However, we find that problematic and decided therefore not follow the suggestions.

Comment: Line 129: please provide a reference for EUCAST

Response: Thank you, we have added the reference (line 149).

Comment: Line 139: please replace „sensitive” with „susceptible”

Response: Thank you, we have changed it (now line 163).

Comment: Table 2: in the first column, please avoid the unnecesary bolding of antimicrobial classes (e.g. aminoglycosides, fenicole ...)

Response: We have removed the bold print of the antibiotic classes.

Comment: Line 127: please provide the selection strategy of the used antimicrobials

Response: We selected antibotics that are frequently used in livestock (line 34-39 in the introduction, see also the cited source). Due to the risk of transmission of antimicrobial resistant isolates to humans (lines 452-454), human-relevant antibiotics were also included in the study, which are not permitted for use in food-producing animals.

Comment: I recommend the inclusion of Tables 4 and 5 in supplementary material files

Response: We followed a reviewer suggestion when we included these details into the main text, and could add from our own perspective that it is easier to have all the information immediately accessible online, without the need for downloads. So, we appreciate the recommendation, but ask kindly to understand that we did not follow.  

Comment: Line 226: „^” – unclear symbol

Response: The declaration of the concentration has been modified (line 254).

Comment: Line 283: „multi-drug resistance” instead of „multiresistance” . Please define this phenomenon in insert the most representative reference for this (e.g. doi: 10.1111/j.1469-0691.2011.03570.x)

Response: Talking about multi-drug resistance here, we don´t operate with fixed expressions like 3MRGN, 4MRGN. So, there is no other definition than the intuitive definition of “simultaneously resistant to several antibiotics”. There exist different definitions for multi-drug-resistance and where it starts (3 ABs, 4 ABs and so on). However, talking about 16fold resistance, these definitions are all met. If we need to add a reference, we would prefer this one, since it expresses the same conception of multi-drug resistance as we have: https://www.ncbi.nlm.nih.gov/pmc/articles/PMC2839888/ (line 316-318).    

Comment: Lines 492-493: „Consumption of raw meat contaminated with (resistant) E. coli presents a health risk to the consumer” – this cannot be considered a conclusion of this work

Response: We agree and deleted that sentence, which was not meant as a conclusion, more as a context for our conclusions. Anyhow, we do not need it here.

Comment: Within the conclusion section the authors must emphasizes the study limitation and further perspectives

Response: We added these aspects, thank you for the hint.

Comment: Reference list: please carefully italicize the scientific name of species throughout the references (e.g. line 528 – for Enterococcus faecium; and 530-531 for Escherichia coli, etc.)

Response: Thanks for the hint, the bibliography has been revised.

Round 2

Reviewer 2 Report

Dear Authors,

Thank you for the corrections and clarifications made. Now the manuscript is much clearer to me.

Best regards

Reviewer 3 Report

All of the raised conc erne have been correctly acknowledged.